# Ionic Silicon Protects Oxidative Damage and Promotes Skeletal Muscle Cell Regeneration

**DOI:** 10.3390/ijms22020497

**Published:** 2021-01-06

**Authors:** Kamal Awad, Neelam Ahuja, Matthew Fiedler, Sara Peper, Zhiying Wang, Pranesh Aswath, Marco Brotto, Venu Varanasi

**Affiliations:** 1Department of Materials Science and Engineering, College of Engineering, University of Texas at Arlington, Arlington, TX 76019, USA; kamal.awad@uta.edu (K.A.); aswath@uta.edu (P.A.); 2Bone-Muscle Research Center, College of Nursing & Health Innovation, University of Texas at Arlington, Arlington, TX 76019, USA; neelam.ahuja@uta.edu (N.A.); matthew.fiedler@uta.edu (M.F.); sara.peper@mavs.uta.edu (S.P.); zhiying.wang@uta.edu (Z.W.); 3Department of Bioengineering, College of Engineering, University of Texas at Arlington, Arlington, TX 76019, USA

**Keywords:** muscle injury, reactive oxygen species, antioxidant, silicon, biomaterials, volumetric muscle loss, tissue regeneration

## Abstract

Volumetric muscle loss injuries overwhelm the endogenous regenerative capacity of skeletal muscle, and the associated oxidative damage can delay regeneration and prolong recovery. This study aimed to investigate the effect of silicon-ions on C2C12 skeletal muscle cells under normal and excessive oxidative stress conditions to gain insights into its role on myogenesis during the early stages of muscle regeneration. In vitro studies indicated that 0.1 mM Si-ions into cell culture media significantly increased cell viability, proliferation, migration, and myotube formation compared to control. Additionally, MyoG, MyoD, Neurturin, and GABA expression were significantly increased with addition of 0.1, 0.5, and 1.0 mM of Si-ion for 1 and 5 days of C2C12 myoblast differentiation. Furthermore, 0.1–2.0 mM Si-ions attenuated the toxic effects of H_2_O_2_ within 24 h resulting in increased cell viability and differentiation. Addition of 1.0 mM of Si-ions significantly aid cell recovery and protected from the toxic effect of 0.4 mM H_2_O_2_ on cell migration. These results suggest that ionic silicon may have a potential effect in unfavorable situations where reactive oxygen species is predominant affecting cell viability, proliferation, migration, and differentiation. Furthermore, this study provides a guide for designing Si-containing biomaterials with desirable Si-ion release for skeletal muscle regeneration.

## 1. Introduction

Skeletal muscle constitutes about 40–50% of total body mass and is responsible for movement of the human body [1]. Although skeletal muscle has remarkable endogenous regenerative capacity, severe traumatic tissue loss greater than 20%, of specific muscle mass, will overwhelm this capacity [2,3] that is known as volumetric muscle loss (VML). The limitations of skeletal muscle to self-repair following severe injuries and the failure to regenerate healthy and functional muscle tissue has opened the door to the investigation of novel alternative approaches for skeletal muscle regeneration. Thus, many tissue engineering approaches have been investigated for VML repair such as autologous minced muscle grafts [4,5,6,7], decellularized extracellular matrix materials (ECM) [8,9], ECMs supplemented with stem or progenitor cells [10,11], and recently inhibition of fibrotic pathways [12]. Furthermore, bioactive scaffolds and cell laden biomaterials [13,14,15] have been proposed to stimulate functional muscle regeneration. These strategies aim to deliver the native ECM with the required cell types (stem or progenitor cells) and growth factors to the injury site, a key step in the regenerative process [16,17]. This process is initiated by the activation of satellite cells, or mononucleated muscle precursor cells, that undergo several proliferative cycles to finally differentiate to form multinucleated myotubes [18,19,20]. Various myogenic transcription factors are expressed within hours of satellite cell activation and regulate the myogenic differentiation process. These factors include myogenic determination protein (MyoD), myogenin (MyoG), and myogenic factor-5 (Myf-5) for cell cycle regulation, and muscle regulatory factor-4 (MRF4) for terminal differentiation. MyoD is a vital gene for myoblast progression and regulation during differentiation to form skeletal muscles [20,21]. Further, there is evidence that increased MyoD acts as a regulator for oxidative metabolism and skeletal muscle biogenesis [21]. The expression of the myogenic marker MyoG is an early indicator of myoblast commitment and differentiation [18,22]. Myokines are molecules released by skeletal muscle and are thought to function as hormone like-molecules, exerting endocrine and paracrine effects on other associated organs and muscle metabolic signaling pathways [23]. Three important myokines include; Neurturin (NRTN) which is a PGC-1α-controlled myokine involved in motor nerve recruitment and neuromuscular junction remodeling, ϒ-aminobutyric acid (GABA) which is the chief inhibitory neurotransmitter in the central nervous system and very important for muscle tonus, and β-aminoisobutyric acid BAIBA, a muscle-derived osteocyte survival factor also implicated in the reduction of insulin resistance and skeletal muscle inflammation [24,25]. NRTN acts via retrograde signaling from muscle to motor neurons, reducing muscle-mass loss, and maintains muscle fiber volume [26]. PGC-1α proteins also have important roles in the biology of skeletal muscle [26]. BAIBA is a known myokine that can protect osteocytes from toxic reactive oxygen species (ROS) and prevent both bone and muscle loss in vivo *due* to unloading [25]. These myokines are thought to be released following activation of the PGC-1a receptor due to stressors, nutrient scarcity, degeneration due to age, injury, and disease and act via retrograde signaling [23,24].

ROS levels at the site of muscle injury critically effects the muscle regeneration process [27,28,29]. At low to moderate levels, ROS stimulates a number of signaling pathways that significantly impact cell cycle, cell migration, survival, apoptosis, and differentiation for tissue healing and muscle repair [28,30]. However, high levels of ROS are often associated with severe traumatic injury, aging, and inherited dystrophy of the skeletal muscles, causing oxidative damage to the skeletal muscle cells, inducing degeneration. This high level of ROS also obstructs the activation of satellite cells affecting natural tissue repair, and ultimately compromise the entire regenerative process [28,31]. Oxidative stress occurs when ROS levels exceed the endogenous cellular antioxidant capacity [32], and as such, exogenous antioxidants are needed to compensate for the excess ROS. Although, experimental evidence has shown that antioxidant supplementations have no effect on muscle function during and after exercise [33], dietary antioxidants have been demonstrated to improve antioxidant enzyme expression and subsequent muscle repair and function in muscle injuries [34,35]. Other investigations have explored administering low doses of antioxidants, leading to reduced oxidative stress, cell cycle and migration stimulation, and enhanced myogenesis [28,36].

Muscle regeneration is a multistep process that starts with the activation of satellite cells, followed by myoblast proliferation and migration, and finally myoblasts fusion and differentiation to produce multinucleated myotubes to replace damaged fibers [28,37]. An ideal muscle treatment or graft should enhance cell functionalities without any adverse effects—complementing endogenous processes rather than bypassing them. In recent years, many bioactive materials have been introduced for musculoskeletal tissue engineering including silicon (Si) and Si-based biomaterials [38,39]. Si-based biomaterials have been shown to promote osteoblast differentiation, extracellular matrix deposition, and bone regeneration [40,41,42]. Silicon carbide has been successfully used as an antithrombogenic coating on vascular stents [43]. Additionally, recent studies have indicated that bioactive silicon nitride, an FDA approved material for spinal intervertebral arthrodesis devices, enhances osteogenic activity and promotes bone ongrowth and ingrowth [44,45]. More recently, bioactive amorphous silicon oxynitride and silicon oxynitrophosphide coatings have induced rapid regeneration of vascularized bone from the antioxidant activity of sustained release silicon-ions [42,46,47]. Furthermore, amorphous silicon oxynitrophosphide coatings enhanced the angiogenic activity of endothelial cells and ionic silicon improved endothelial cell survival under toxic ROS conditions via enhanced angiogenic marker expression and antioxidant activity [40,42,48]. Additionally, research conducted by Monte et al. suggests that Si ions at a concentration of 0.5 mM mitigate the toxic effects of ROS by catalyzing the conversion of superoxide to peroxide [21]. The addition of laponite nanosilicates to 3D printed hydrogels supported osteoblast function and enhanced mechanical strength [39,49]. Recent evidence indicates that C2C12 myoblasts cells cultured on a silicon substrate maintained normal biological activities [50]. Research has shown that silica nanoparticles have a beneficial effect on myoblast fusion in C2C12 skeletal muscle cells [18]. Additionally, our preliminary data indicates that micro-patterned silicon oxynitride enhanced adhesion, growth, and myotube and axon alignment of muscle and nerve cells, and that ionic silicon elutes from the material and into solution under physiologic conditions [51,52,53].

Yet, the effect of Si-ions on skeletal muscle cell activity, such as proliferation, migration, differentiation, and myogenic biomarker expression, has not been explored. Thus, the effect of ionic Si released from Si-based biomaterials needs to be understood in skeletal muscle cells for the optimization of cell-based therapies and tissue engineering applications. In this study, we aim to understand the effect of Si-ions on C2C12 skeletal muscle cells under normal (no added ROS) and toxic ROS conditions to gain insight into its role on myogenesis during the early stages of muscle regeneration. First, C2C12 cell viability, proliferation, migration, and differentiation were studied following the addition of three different concentrations of Si-ions to cell culture medium to determine the optimal Si-ions dose for C2C12 cells. Second, we studied the effect of Si-ions on myogenic gene expression (i.e., MyoD and MyoG) and myokines (i.e., Neurturin, GABA, and BAIBA) expression. Third, we studied the effects of ROS on C2C12 cell viability by culturing cells in five different concentrations of H_2_O_2_ to determine the minimum H_2_O_2_ concentration that induces a significant oxidative damage. Then, four different concentrations of Si-ions were added to culture medium to study the effect of Si-ions on C2C12 cells under toxic oxidative stress condition (0.4 mM H_2_O_2_).

## 2. Results

### 2.1. Effects of Si-Ions Dose on C2C12 Myoblasts

#### 2.1.1. Si-Ions Enhance C2C12 Cell Viability

The effect of Si-ions on C2C12 myoblast cell viability was tested by using three different concentrations of Si-ions at two time points (6 and 24 h) as shown in Figure 1. Under normal conditions, all Si-ions concentrations showed no cytotoxic effects on C2C12 cells. After 6 h, all Si-ion groups displayed higher cell numbers compared to the positive control (GM), but no significant difference was observed as shown in Figure 1A. After 24 h, 0.1 mM of Si in GM significantly increased cell viability compared to the control (* *p* < 0.05, *n* = 3 per group) as indicated by the MTS-assay results, expressed as optical density (OD) in Figure 1A. Furthermore, live/dead stain confirmed the addition of Si-ions increased cell viability as indicated by the green stain of live cells after 24 h (Figure 1D). The number of cells counted for each group based on live stain fluorescent images (6 FOVs) confirmed that 0.1 mM of Si-ions significantly increased cell numbers compared to the control (* *p* < 0.05, *n* = 3 per group), congruent with MTS results. Finally, following normalization of cell counts to the control it was determined that there was a 1.4-fold increase in cell counts with 0.1 mM Si.

#### 2.1.2. Si-Ions Promote C2C12 Cell Proliferation

Figure 2 presents the effect of Si-ions on C2C12 cell proliferation following 3 days cell culture under normal conditions. Following 1, 2, and 3 days of C2C12 proliferation in Si ions it was determined that cell proliferation increased in all Si-ion groups compared to the control at day 1, and there was a significant increase in cell proliferation in 0.1 mM Si-ion samples following 1 and 3 days (* *p* < 0.05) based on the OD results of the MTS assay. Figure 2B presents the fluorescent images of C2C12 cells stained with live/dead assay kit for live (green) and dead (red) cells following 2 days proliferation. Almost no dead cells were observed for the control and Si-ion groups (*p* > 0.5). An increased number of cells during proliferation were seen in Si ion samples compared to controls was confirmed by counting the 9 FOVs for each well. There was a significant increase in cell number in 0.1 mM Si-ion groups at day 1 (* *p* < 0.05) and day 2 (*** *p* < 0.001). Furthermore, the area covered by live C2C12 cells was calculated using the fluorescent images as shown in Figure 2D. Percentage of cell area coverage confirmed that the addition of Si-ions significantly increased the area covered by cells during proliferation.

#### 2.1.3. Si-Ions Promote C2C12 Cell Differentiation

To determine the effect of Si-ions on myogenic differentiation, C2C12 cells were cultured with three different concentrations of Si-ions for 4 and 7 days. After each timepoint, the mature myocytes/myotubes were stained with MHC antibody and DAPI counterstain as shown in Figure 3. FI calculations indicated that 0.1 mM Si-ions significantly increased myotube formation compared to the control with FI (%) = 42 ± 3.3 and 31 ± 4 at day 4 and FI = 60 ± 3.2 and 50 ± 2.4, respectively, at day 7 (*** *p* < 0.001) as shown in Figure 3A. Additionally, 0.5 mM and 1.0 mM Si-ions significantly increased the FI compared to the control at day 7, (** *p* < 0.01 and * *p* < 0.05, respectively). Overall, FI was significantly increased from day 4 to day 7 for all samples including the control (*** *p* < 0.001). Total number of cells counted from 9 FOVs of DAPI stained nuclei indicated that all Si concentrations significantly increased the total number of cells at day 4 compared to the control with optimal concentration of 0.1 mM Si-ions, *** *p* < 0.001 (Figure 3B). Furthermore, area covered by myotubes was calculated based on MHC staining at 4 and 7 days. The calculated myotubes area (%) indicated that all Si concentrations increased the total area of myotubes per FOV compared to the control, however, only 0.1 mM Si-ions presented a significant difference at day 4 (34.3 ± 1.2%) and day 7 (48.9± 2.4%) when compared to the control (25.3 ± 3.6% and 40.6 ± 4.8% for day 4 and 7, respectively) as shown in Figure 3C. It is important to mention that there was no significant difference between the total number of cells from day 4 to 7, while the FI and the myotube area % significantly increased from day 4 to day 7 for all groups including the control.

#### 2.1.4. Si-Ions Enhance MyoG and MyoD Gene Expression

Expression of two key regulatory genes of myogenesis related to C2C12 myoblast differentiation were detected using qRT-PCR analysis following 1 and 5 days of differentiation. MyoG is an important myogenic regulatory factor expressed to mark cell commitment to differentiation [21]. MyoD is a vital gene for myoblast progression during differentiation to form skeletal muscles [22]. Figure 4 shows relative mRNA expression of MyoG and MyoD following 1 and 5 days of differentiation. Figure 4B presents the mRNA expression of MyoG at day 1 and 5 of C2C12 differentiation. All Si-ion concentrations significantly increased MyoG expression compared to control at day 1 and 5. Following day 1 of differentiation MyoG expression increased ~1.3-fold in both 0.1 mM and 0.5 mM Si-ion groups (** *p* < 0.01, * *p* < 0.05, respectively) then further increased following 5 days of differentiation to ~1.8-fold difference for 0.1 mM Si-ions and ~1.5-fold difference for 0.5 mM Si-ions (*** *p* < 0.001, ** *p* < 0.01, respectively) compared to control. For the high concentration 1.0 mM Si-ion group, there was a significant increase by ~2-fold in MyoG expression at day 1 which decreased to a ~1.8-fold difference compared to the control by day 5 of differentiation (*** *p* < 0.001). The qRT-PCR results indicated that MyoD expression was significantly increased in 0.1, 0.5 and 1.0 mM of Si-ion samples following 1 and 5 days of C2C12 myoblast differentiation as shown in Figure 4A (*** *p* < 0.001). A 2.5-fold increase in MyoD expression for 0.1 mM Si-ion samples and a 3-fold increase in 1.0 mM Si-ion samples was observed compared to control at day 1. There was a 3-fold difference in MyoD expression in the 0.1 mM Si-ion group and only a 2.8-fold difference in the 1.0 mM Si-ion group following 5 days of differentiation. Finally, there was a 2.5-fold increase in MyoD expression in the 0.5 mM Si-ion group following 1 day of differentiation that decreased to a 2-fold difference by day 5.

#### 2.1.5. Si-Ions Enhance Myokines Expression

Collected media were tested to determine the effect of Si-ions on C2C12 biomarker expression following 5 days of differentiation under normal conditions. NRTN expression following 3 and 5 days of differentiation was determined by ELISA assay. NRTN concentration (ng/mL) was increased in Si-ion samples at day 3 compared to the control (Figure 5C). NRTN expression significantly increased in the 0.5 mM Si-ion group compared to the control following 3 days of differentiation (** *p* < 0.01). NRTN expression following 5 days of differentiation indicated that NRTN concentration decreased as differentiation time increased from 3 to 5 days. Although 0.5 mM and 1.0 mM of Si-ions reported the highest NRTN concentration at day 3 of differentiation, 0.5 mM and 1.0 mM Si-ions reported the lowest NRTN concentration by day 5 of differentiation. The initial increase followed by reduction in NRTN levels may indicate that NRTN secretion was increased, however, the inability to recruit motor neurites via retrograde signaling in vitro resulted in a quick reduction in NRTN concentration.

The collected media was also tested to study the effect of Si-ions on aminobutyric acid expression during C2C12 differentiation. Although the analysis was performed for all isomers of aminobutyric acid (AABA, BABA, and GABA), only GABA and β-aminoisobutyric acid (D-BAIBA) were detected in DM as indicated in Figure 5. At day 1 of differentiation, GABA expression was increased for all Si-ion groups as compared to the control but there was only a significant increase in GABA ~0.42 ± 0.02 µM in the 0.1 mM Si-ion group (* *p* < 0.05). At 3 and 5 days of differentiation, GABA concentration was significantly reduced for all Si-ions groups and the control compared to day 1 concentrations. GABA concentration was higher in the 0.5 mM Si-ion group compared to the other groups at day 3. Overall, the GABA concentration reduced dramatically after the first day of differentiation indicating that GABA expression decreases as differentiation time increases. Additionally, there was no significant difference between the control and Si-ions-media on GABA expression at day 5. D-BAIBA concentration was increased in the 0.5 mM Si-ion group (~0.14 ± 0.02 µM) compared to the control (~0.12 ± 0.01 µM) at day 1 of differentiation but no significant difference was observed (Figure 5B). While the D-BAIBA concentration was higher for all Si-ions groups compared to the control on day 3, only the 1.0 mM Si-ion group was significantly different (* *p* < 0.05). D-BAIBA concentrations reduced dramatically at day 3 and 5 compared to day 1 of differentiation, indicating that D-BAIBA concentration decreases as differentiation time increases. The initial increase followed by reduction in both GABA and D-BAIBA levels again suggest that myokine secretion was increased, however, the lack of retrograde signaling in vitro resulted in a quick reduction in myokine concentration.

### 2.2. Effects of Different H_2_O_2_ Concentrations and Si-Ions on C2C12 Cells

#### 2.2.1. H_2_O_2_ Impairs C2C12 Myoblast Cell Viability

The effect of H_2_O_2_ on C2C12 myoblast cell viability was tested using five different concentrations (0.2, 0.4, 0.6, 0.8, and 1.0 mM H_2_O_2_) compared to the normal GM as a positive control as shown in Figure 6. MTS and live/dead assays were used to determine cell viability at 6 and 24 h following the addition of H_2_O_2_. All H_2_O_2_ concentrations significantly decreased cell viability at 6 and 24 h compared to the control (*** *p* < 0.001, ** *p* < 0.01, *n* = 4 per group) as shown in Figure 6A. Live/dead staining confirmed the results obtained from the MTS assay by showing the gradual decrease in the total number of live cells (green) and the increase in dead cells (red) as the H_2_O_2_ concentration increased from 0.2 to 0.6 mM as shown in Figure 6D. No live cells were observed for 0.8 and 1.0 mM H_2_O_2_ while the number of dead cells increased. The fluorescent images were further used to count the number of live cells per 6 FOVs for each group (Figure 6B) and the cell number was normalized to the control as shown in Figure 6C. The cell number was significantly reduced in all H_2_O_2_ groups compared to the control, while 0.4 mM H_2_O_2_ was the minimum concentration for inducing a high significant difference in cell number (*** *p* < 0.001). The normalized cell number indicated that 0.2 mM H_2_O_2_ decreased the number of viable cells by 37% (** *p* < 0.01) while 0.4 mM H_2_O_2_ decreased viable cells by 70% at 24 h (*** *p* < 0.001). These results confirmed the results obtained by the MTS assay.

#### 2.2.2. Si-Ions Protect C2C12 Myoblast Cells against Oxidative Damage

As presented above, 0.4 mM H_2_O_2_ was the minimum concentration to induce a significant effect (*** *p* < 0.001, R^2^ = 0.99) on C2C12 cell viability. To study the effect of Si-ions on C2C12 under toxic oxidative stress conditions, 0.4 mM H_2_O_2_ was added to GM to induce ROS related damage, then four different concentrations of Si-ions were added to determine the optimal concentration that attenuated the effect of toxic oxidative stress. Cell viability was measured by MTS assay at 6 and 24 h as shown in Figure 7A, and the five groups were compared to the control. The 0.4 mM H_2_O_2_ group (i.e., 0.0 mM Si) presented a significant decrease in cell viability at 6 and 24 h compared to the control. At 6 h, there was no significant difference in cell viability for low concentrations of Si-ions (0.1 and 0.5 mM), while higher concentrations of Si-ions (1.0 and 2.0 mM) attenuated the effect of toxic H_2_O_2_ as indicated by a significant increase in cell viability (*** *p* < 0.001, R2 = 0.98) compared to the 0.4 mM H_2_O_2_ group. At 24 h, all Si-ion concentrations attenuated the toxic effects of H_2_O_2_ resulting in increased cell viability compared to the 0.4 mM H_2_O_2_ group (*** *p* < 0.001, R^2^ = 0.99).

Figure 7C presents the fluorescent images after 24 h for all groups. The number of live cells (green) increased in all Si-ions groups compared to the 0.4 mM H_2_O_2_ group which presented a higher number of dead cells (red). The fluorescent images were used to count the number of live cells per 6 FOVs for each group, then cell number was normalized to the 0.4 mM H_2_O_2_ group as shown in Figure 7B. The normalized cell counts indicated that live cell number was significantly increased in all Si-ion groups, congruent with the results obtained from the MTS assay. Importantly, 0.5 mM Si-ions was the optimal concentration of Si-ions that could attenuate the effect of toxic H_2_O_2_ as indicated by a significant increase in cell viability (*** *p* < 0.001).

#### 2.2.3. Si-Ions Enhance Cell Differentiation under Toxic Oxidative Stress

After determining the optimal concentration of Si-ions for attenuating the toxic effects of oxidative stress on C2C12 myoblast cells, a cell differentiation experiment was performed to study the effect of Si-ions on cell differentiation under toxic H_2_O_2_. 

DM was used as a control to compare 0.5 mM Si-ions, 0.4 mM H_2_O_2_, and 0.4 mM H_2_O_2_ + 0.5 mM Si-ion groups as shown in Figure 8. The fluorescent images of DAPI-stained nuclei (blue) and myosin heavy chain antibody (MHC, green)-stained myocytes/myotubes after 4 days of differentiation are presented in Figure 8C. FI for control and 0.5 mM Si-ions groups was significantly higher than the other groups (*** *p* < 0.001). FI was increased (22 ± 5) in the 0.5 mM Si-ion group under oxidative stress compared to 0.4 mM H_2_O_2_ (18 ± 6), but no significant difference was observed. Based on the MHC-stained myotubes, the myotube area coverage (%) was calculated per 9 FOVs as shown in Figure 8B. Addition of 0.5 mM of Si-ions resulted in a significant increase in myotube coverage area from 9.5± 2.4% for the 0.4 mM H_2_O_2_ group to 12.2 ± 1.6% for the 0.4 mM H_2_O_2_ + 0.5 mM Si group (** *p* < 0.01). Figure 8D, E shows the relative mRNA expression of MyoG and MyoD following 1 and 5 days of differentiation under ROS condition. It is clear that 0.4 mM H_2_O_2_ significantly decreases the MyoG (### *p* < 0.001) and MyoD (# *p* < 0.05) expression at day 1 of differentiation. Although the combined group (0.4 mM H_2_O_2_ + 0.5 mM Si) presented a significant decrease in MyoG (### *p* < 0.001) expression, no significant change was observed for MyoD expression at day 1. Furthermore, comparing 0.4 mM H_2_O_2_ group to the Si treated group (0.4 mM H_2_O_2_ + 0.5 mM Si) revealed that Si addition significantly increased the MyoD and MyoG expression (* *p* < 0.05) under ROS condition at day 1 of differentiation. On the other hand, day 5 results of MyoG and MyoD did not reveal any significant difference compared to the control.

#### 2.2.4. Si-Ions Enhance Antioxidant Marker Expression in the Presence of Toxic Oxidative Stress

To determine the effects of Si-ions on antioxidant activity in the presence of toxic oxidative stress, C2C12 cells were differentiated in media containing 0.5 mM Si-ions only, 0.4 mM H_2_O_2_ only, 0.5 mM Si + 0.4 mM H_2_O_2_, and no treatment group as a control. After 1 and 5 days of differentiation, real time qRT-PCR was used to quantify the relative concentrations of NRF-2 and SOD-1 mRNA in all 4 groups (Figure 9). At normal condition, Si treatment did not show any significant effect on NRF-2 or SOD-1 expression compared to the control, while 0.4 mM H_2_O_2_ has increased the SOD-1 and NRF-2 expression with a significant increase only in NRF-2 expression at day 1 and 5 (Figure 9A). For the combined group (0.5 mM Si + 0.4 mM H_2_O_2_), there was a significant increase in NRF-2 (1.7 ± 0.1, *p* = 0.00002) and SOD-1 (1.35 ± 0.06, *p* = 0.04) at day 5 compared to the control.

#### 2.2.5. Si-Ions Enhance Wound Healing and Cell Migration Rate under Toxic Oxidative Stress

Scratch-wound healing assay was performed to compare cell migration parameters under different conditions. It is commonly used to measure cells’ migration parameters such as speed, persistence, and polarity [54]. In this study, we compared myoblast cells’ migration rate, referred to as the wound healing rate, under four different conditions: 0.1–1.0 mM Si-ions, 0.4–0.8 mM H_2_O_2_, 0.4 mM H_2_O_2_ + 1.0 mM Si-ions, and the control as shown in Figure 10. The cells’ migration rate under Si-ions compared to the control is shown in Figure 10A. It is noted that 0.5–1.0 mM Si-ions slightly increased the migration rate, but no significant difference was observed. Figure 10B shows the migration rate under three toxic H_2_O_2_ concentrations (0.4, 0.6, and 0.8 mM). All used H_2_O_2_ concentrations significantly decreased the migration rate (*** *p* < 0.001) compared to the control group as shown in Figure 10B. Addition of 1.0 mM Si-ions to the 0.4 mM H_2_O_2_ conditioned media attenuated the toxic effect of H_2_O_2_ and significantly increased the cells’ migration rate compared to the non-Si-ions treatment (** *p* < 0.01) as shown in Figure 10C. Furthermore, no significant difference was observed between the control group (rate = 0.033 ± 0.001 mm^2^/h) and the H_2_O_2_ + Si group (rate = 0.031 ± 0.002 mm^2^/h), while the H_2_O_2_ alone presented a significant decrease in the migration rate (rate = 0.025 ± 0.001 mm^2^/h) compared to the control (*** *p* < 0.001). Figure 10D shows the phase contrast images (10×, scale bar = 100 μm) of wound/scratch area at different time points (0, 12, 24, and 28 h). Migration rate of each group is reported in the inserted table in Figure 10.

## 3. Discussion

Our analyses revealed the novel effect of ionic Si-ions on C2C12 myoblasts under normal and toxic oxidative stress conditions. After confirming the positive effect of Si on myoblast functionalities, we studied its effect on C2C12 myoblasts under toxic oxidative stress conditions produced by H_2_O_2_ as a source of ROS. Our key results were that ionic Si enhanced the C2C12 viability, proliferation, differentiation, and myogenic gene and marker expression. Importantly, our results indicate that Si-ions at a concentration of 1.0 mM accelerate muscle wound healing by increasing the cell migration rate protecting it from the toxic oxidative damage compared to a positive control. Previous studies had indicated that antioxidants can improve muscle mass recovery [36], enhance viability by reducing muscle cell death [55], and promote the early stages of differentiation by inducing the expression of differentiation markers [28,56,57]. In line with this, our results indicate that Si-ions at a concentration of 0.1 mM can enhance C2C12 cell viability, proliferation, differentiation, and myogenic marker expression, such as MyoG and MyoD, analogous to the antioxidant effect of Si-ions on osteogenesis [41,47,48].

Cell viability and proliferation experiments confirmed that Si-ions at concentrations of 0.1–1.0 mM supports myoblast cell growth, and 0.1 mM Si-ions is the optimal concentration to enhance C2C12 cell viability and proliferation. At early timepoints, all Si-ion concentrations resulted in higher cell numbers compared to the positive control. After 24 h, 0.1 mM of Si ions in GM significantly increased cell viability compared to the control, congruent with the results of MTS, live/dead assay, and the fluorescent image cell counting. Furthermore, all Si-ion concentrations significantly enhanced C2C12 fusion at an early stage of differentiation (day 4) compared to the control. Specifically, 0.1 mM Si-ions significantly increased myoblast fusion as indicated by FI calculations. Additionally, the size and length of the formed myotubes were more pronounced for the Si-ions groups compared to the control, determined by calculating the area of formed myotubes. It is important to note that the increase of FI and myotube area from day 4 to day 7 of differentiation occurred despite no change in the total number of cells, counted from DAPI stain, from day 4 to 7 indicating the successful differentiation of C2C12 myoblasts.

This enhancement of C2C12 differentiation can be attributed to the early expression of myogenic markers such as MyoD, which was overexpressed almost 3-fold at day 1 of differentiation compared to control. The effect of Si-ions can be clearly seen from MyoD and MyoG overexpression compared to the Si-free control. MyoG was expressed at a later stage by day 5 to mark the commitment to differentiation. While MyoG expression was significantly increased by 2-fold in the high concentration (1.0 mM) Si-ion group at day 1, an almost 2-fold increase in MyoG expression was observed in both high and low concentration (0.1- and 1.0-mM Si-ions) groups compared to the control at day 5. The early expression of MyoG at day 1 can be attributed to the high dose of Si-ions. Overall, the enhancement in C2C12 cell functionalities such as proliferation and differentiation can be attributed to the antioxidant effect of Si-ions that promotes the early stage of differentiation as indicated by the myogenic marker’s expression. The results of this study indicate that certain concentrations of ionic Si promote maximal expression of myogenic biomarkers.

Our results indicated that addition of Si-ions to the differentiation media can increase NRTN, BAIBA, and GABA expression at day 1 of differentiation as compared to the control without Si-ions. Although all Si-ion concentrations increased myokine expression at day 1, myokine expression was drastically reduced as differentiation time increased from day 1 to day 5 for all samples including the control. The initial increase in GABA and D-BAIBA levels followed by reduction in both suggest that myokine secretion increased, however, the lack of retrograde signaling in vitro resulted in a quick reduction in myokine concentration. On the other hand, it has been well established that ROS has a dose-dependent effect on assisting or hindering tissue regeneration. Low to moderate levels of ROS can enhance regeneration by inducing cell migration and differentiation [29,30]. While high levels of ROS can delay the regeneration process and prolong hospital stays, a common scenario in severe muscle injuries [31,58,59]. Thus, antioxidant supplements have been used to attenuate the harmful effects of high level of ROS in severe injuries [28,58,60].

Excessive oxidative stress is a major contributor for various skeletal muscle tissue pathologies including prolonged severe traumatic injuries, sarcopenia, degenerative diseases and also other muscular atrophies [61]. Severe traumatic injuries and degenerative muscle loss can be due to the presence of additional reactive species and the reduction of the cells capacity to remove it obstructs the cellular repair process along with obstruction in activation of the satellite cells to induce muscle regeneration. Oxidative stresses at higher amounts also creates an imbalance impairing the functionality of proteins and cellular structures, also high concentration of myoglobin in skeletal muscles making it more sensitive to free radical induced oxidative damage especially in chronic degenerative cases [61,62].

Thus, our second aim was to study the effect of Si-ions on C2C12 activity under toxic oxidative stress. Our objectives were to determine the minimal concentration of H_2_O_2_ needed to induce oxidative damage on C2C12 cells, then determine the optimal Si-ion concentration to counteract or attenuate the effect of H_2_O_2_. We performed cell viability studies with five different concentrations of H_2_O_2_ to determine the minimal concentration of ROS for inducing significant cell death. All used H_2_O_2_ concentrations had a negative impact on cell viability, but 0.4 mM was the minimum concentration for inducing a strong significant difference (*** *p* < 0.001). Results of MTS assay and cell counting indicated that 0.2 mM H_2_O_2_ decreased the number of viable cells by 37%, while 0.4 mM H_2_O_2_ decreased the number of live cells by 70% at 24 h. Then, 0.4 mM H_2_O_2_ was used to study the effect of Si-ions in a toxic oxidative stress environment, and four different concentrations of Si-ions were tested. The results indicated that high concentrations of Si-ions (1.0–2.0 mM) can attenuate the effect of toxic ROS after only 6 h, while the low concentrations of Si-ions induced significant enhancement of cell viability after 24 h compared to control. Thus, we concluded that 0.5 mM Si-ions is the optimal concentration of Si-ions for attenuating the effect of toxic H_2_O_2_ based on a significant increase in cell viability. Furthermore, after determining the effective concentration of H_2_O_2_ (0.4 mM) and the effective concentration of Si-ions (0.5 mM), a cell differentiation study was performed to investigate if Si-ions have any effect on fusion. FI was increased in the 0.5 mM Si-ion group (22 ± 5) under oxidative stress conditions compared to 0.4 mM H_2_O_2_ (18 ± 6), but no significant difference was observed. Previous studies indicated that antioxidants reduce the deleterious effects of H_2_O_2_ on cell migration but not cell fusion [28]. Although the FI results are in agreement with this previously reported effect, here we found that 0.5 mM of Si-ions significantly increased the myotube coverage area from 9.5 ± 2.4% for the 0.4 mM H_2_O_2_ group to 12.2 ± 1.6% for the 0.4 mM H_2_O_2_ + 0.5 mM Si group. This indicates that measuring the myotubes area combined with FI calculations is a more accurate method to assess cell differentiation. This enhancement in cell differentiation under ROS condition with Si treatment was further confirmed by MyoD and MyoG gene expression analysis.

Our last observation was that 0.5–0.1 mM Si-ions increased the cell’s migration rate under normal conditions compared to the control. Furthermore, 1.0 mM Si-ions significantly enhanced muscle wound healing by increasing the cells’ migration rate under toxic oxidative stress condition. In a previous study, 0.1 mM H_2_O_2_ increased C2C12 cell migration rate while the high concentration of 0.5–1 mM hindered cell mobility [28]. In line with this, our data indicated that 0.4–0.8 mM H_2_O_2_ significantly decrease the cells’ migration rate that usually results in poor wound healing and prolonged healing time under severe muscle injuries. Adding 1.0 mM Si to the media with H_2_O_2_ attenuated the toxic effect and enhanced the healing rate under these conditions. It is important to mention the advantages of using the live dynamic imaging technique in such wound-healing assays (See Appendix A). Firstly, it allows to capture cells’ migration images at the same exact position over long periods of time until complete wound/scratch healing. Secondly and as, we noted, the scratches closed in 27, 27.8, 35, and 28.7 h, respectively, a small interval between completion times that may have been overlooked with less frequent imaging. Additionally, the healing rates based on 6-h readings were slightly different than the obtained healing rates based on the live dynamic imaging technique. Overall, we can conclude that 1.0 mM Si-ions attenuate the effect of H_2_O_2_ and enhance cells’ migration rate compared to the control.

Additionally, the results of qRT-PCR assays suggest that Si may upregulate the expression of SOD-1 when in the presence of oxidative stress, however, the same effect was not observed under normal conditions or oxidative stress conditions in the absence of Si-ions. Although 0.4 mM H_2_O_2_ group presented a significant increase in NRF-2 expression (1.4-fold change), the expression was more pronounced under Si treatment with 1.7 fold change. This suggests that Si-ions may play a role in mitigating the effects of excessive reactive oxidative species by upregulating antioxidant markers. H_2_O_2_ can react with metal ions in the cells to produce hydroxyl radical which is one of the most reactive species in the biological systems [63]. High levels of ROS with imbalanced antioxidants expression usually generates an oxidative stress that leads to degenerative changes to muscle with all of the characteristics of a muscular dystrophy [61,62,63,64]. In this regard, Si treatment could be a beneficial approach for upregulation of SOD-1 in some muscle dystrophies related SOD deficiencies. Furthermore, the cells’ migration results combined with cell viability and differentiation indicate that ionic Si may have a potential role in unfavorable situations where ROS are predominant, to protect from oxidative damage and inducing muscle cell regeneration in case of traumatic muscle injury and delaying the degenerative process of the muscle improving the life expectancy in degenerative muscle diseases.

## 4. Materials and Methods

### 4.1. Experimental Study Design

To study the effect of Si-ions on C2C12 skeletal muscles under normal and oxidative stress conditions, a series of 2D cell culture experiments were performed. To examine the effects of Si-ions on C2C12 cells under normal conditions, cells were cultured in normal media and in media containing Si-ions at concentrations of 0.1 mM, 0.5 mM, and 1.0 mM. Si ions were obtained using sodium metasilicate powder (Na_2_SiO_3_) [48,65,66,67]. Cell viability and proliferation assays were performed on cells cultured in growth media with Si-ions while cell morphology studies were carried out in differentiation media with Si-ions. Cell differentiation media was collected periodically throughout differentiation for biomarker and gene expression analysis. Following cell differentiation, cells were fixed, immunohistochemically stained, and imaged under fluorescence to determine the optimal concentration of Si-ions for myogenic differentiation. Next, cell viability and proliferation assays were conducted on C2C12s cultured in growth medium (GM) containing different concentrations of hydrogen peroxide to determine the concentration of hydrogen peroxide required to induce oxidative damage (significant cell death). The optimal concentration of Si-ions for mitigating toxic ROS activity was then determined by differentiating C2C12 cells in media containing 0.4 mM hydrogen peroxide and Si-ions at concentrations of 0.1 mM, 0.5 mM, 1.0 mM, and 2.0 mM and Si-ions free controls. Following optimization of Si-ions concentration, a final cell culture study was conducted to determine the effects of optimized Si-ions concentrations on cell viability, proliferation, morphology, healing rate, and gene and biomarker expression under the determined concentration of H_2_O_2_. The cell viability, proliferation, and morphology were determined in the same manner as described above. Media was collected periodically for biomarker and gene expression analysis using qPCR and ELISA. Finally, cell migration was evaluated using a scratch test [48,68] to determine effects on healing rate.

### 4.2. Materials

Sodium metasilicate powder (Na_2_SiO_3_, MW: 122.06 g/mol, Sigma-Aldrich Co., St. Louis, MO, USA) and hydrogen peroxide (H_2_O_2_, 30% *w*/*w*, Sigma-Aldrich Co., St. Louis, MO, USA) were used as sources of Si-ions and ROS, respectively. Dulbecco’s Modified Eagle’s Medium ((DMEM-1×) with 4.5 g/L glucose, L-glutamine, and sodium pyruvate), phosphate-buffered saline (PBS-1×), penicillin-streptomycin (P/S) 10,000 U/mL each, and trypsin EDTA-1× solution were purchased from Mediatech Inc. (Manassas, VA, USA). Fetal bovine serum (FBS) and horse serum (HS) were obtained from Thermo Fischer Scientific Inc. (Waltham, MA, USA). C2C12 mouse myoblast skeletal muscle cell lines were obtained from The American Type Culture Collection (ATCC) (Manassas, VA, USA) after authentication and testing for mycoplasma contamination. CellTiter 96^®^ AQueous One Solution Cell Proliferation Assay (MTS) was obtained from Promega (Madison, WI, USA). Diamidino-2-phenylindole (DAPI), a blue, fluorescent nucleic acid stain, was purchased from Sigma-Aldrich Co., (St. Louis, MO, USA). Human Myosin Heavy Chain (MHC) fluorescein-conjugated antibody was purchased from R&D Systems, Inc. (McKinley Place, MN, USA). Invitrogen™ LIVE/DEAD™ Viability/Cytotoxicity Kit was purchased from Thermo Fischer Scientific Inc. (Waltham, MA, USA). Mouse Neurturin (NRTN) ELISA Kit (Catalog Number. CSB-EL016095MO) was purchased from Biomatik USA, LLC (Wilmington, DE, USA). (S)-α-aminobutyric acid (L-AABA) and (R)-α-aminobutyric acid (D-AABA) were purchased from Thermo Fisher Scientific Inc. (Waltham, MA, USA). (S)-β-aminoisobutyric acid (L-BAIBA) and (R)-β-aminoisobutyric acid (D-BAIBA) were purchased from Adipogen Corp. (San Diego, CA, USA), and GABA was purchased from Sigma-Aldrich Co., (St. Louis, MO, USA). For quantitative real-time polymerase chain reaction (qRT-PCR): the RNeasy Mini Kit was obtained from Qiagen (Valencia, CA, USA), the reverse transcription system was obtained from Promega (Madison, WI, USA), and the PCR primers for the targeted genes, MyoD and MyoG, were obtained from Thermo Fischer Scientific Inc. (Waltham, MA, USA).

### 4.3. Preparation of Si-Ions and Hydrogen Peroxide Solutions

Na_2_SiO_3_ was used as a source of Si-ions. To prepare 100 mM of Si, 610.3 mg of Na_2_SiO_3_ was dissolved in 50 mL of sterile DI water then filtered using a nylon syringe filter (33 mm, 0.2 μm,) followed by a serial dilution, the desired concentrations of Si were obtained (0.1, 0.5, and 1.0 mM) in C2C12 myoblast cell growth media. Hydrogen peroxide is a form of ROS generated when the enzyme superoxide dismutase scavenges superoxide anions [69], thus H_2_O_2_ was used as a source of ROS and inducing oxidative damage. 100 mM-H_2_O_2_ solution was prepared in 10 mL sterile DI water, filtered, and serially diluted in sterile DI water until the desired concentrations of H_2_O_2_ (0.2, 0.4, 0.6, 0.8, and 1.0 mM) were reached in C2C12 cell growth media.

### 4.4. Cell Culture Studies

C2C12 myoblast is an immortal cell line that are easy to obtain and passage. Typically, C2C12 myoblasts differentiate rapidly under normal, Si-free, conditions, forming contractile myotubes and producing proteins with characteristics similar to endogenous muscle tissue. Using C2C12 myoblasts provides a more stable environment with fewer confounding variables than the normal physiologic environment of human primary cells [70]. Thus, C2C12 myoblasts were cultured using well-established and previously published protocols [71,72,73]. Briefly, C2C12 cells were cultured in growth medium (GM), GM consisting of DMEM/high glucose + 10% FBS + 100 U/mL P/S, in 75-cm^2^ Corning cell culture flasks with canted neck and vented caps under normal conditions (37 °C, 5% CO_2_) maintaining cell density at 40–70% confluency. Cells in GM were maintained at 40–70% confluency, allowing them to proliferate but not to differentiate into myotubes. GM was changed every 48 h. For differentiation studies, after reaching 75% confluency, C2C12 myoblast cells were cultured in differentiation medium (DM), DMEM/high glucose + 2% HS + 100 U/mL P/S, which was changed every 48 h for 5–7 days until fully developed myotubes were formed.

#### 4.4.1. Effects of Different Ionic Si-Ions Concentrations on C2C12 Cells Functional Capacity

To study the effect of Si-ions on C2C12 myoblast viability and proliferation, 2.5 × 10^4^ cells/well were seeded in 24-well plates with 500 µL of GM and incubated as described above. Three different concentrations of Si-ions in GM were tested compared to the normal GM as control. Sample size of *n* = 6 was used for each group; GM, GM + 0.1 mM Si, GM + 0.5 mM Si, and GM + 1.0 mM Si-ions. These different concentrations of Si-ions were added to the media during seeding, then cells were incubated for 6 and 24 h for cell viability experiments and for 1, 2, and 3 days for cell proliferation studies. After each time point, MTS assays were performed (*n* = 3 for each group) using the CellTiter 96 AQueous One Solution Cell Proliferation Assay kit according to the manufacturer’s protocol. Briefly, MTS solution was prepared by adding 40 µL of CellTiter 96 reagent to 200 µL of GM for each sample to be tested. At each time point GM was removed from each sample and 240 µL of the MTS solution was added to each well of the 24-well plate. After 3 h, 100 µL samples were collected from each well twice and transferred into two separate wells of a new 96 well-plate. The colorimetric absorbance was determined using a microplate reader (SpectraMax^®^ i3, Molecular Devices, CA, USA) at 490 nm. The remaining *n* = 3 for each group each time point was used for the live/dead assay using Invitrogen LIVE/DEAD Viability/Cytotoxicity Kit according to the manufacturer’s protocol. Briefly, 200 µL of live/dead solution was added to each well and incubated for 30 min. Fluorescent images were then taken using a DMi8 inverted Leica microscope (Leica Microsystems Inc., IL, USA), with green staining for live cells and red staining for dead cells. Three fluorescent images (20× field of view (FOV)) were taken for each well to compare the number of live and dead cells in samples. The number of live and dead cells in fluorescent images were counted using ImageJ software (NIH).

To study the effect of Si-ion concentration on cell morphology and differentiation capacity, C2C12 cells were cultured for 4 and 7 days in DM containing 3 different concentrations of Si-ions compared to normal DM as control. For all differentiation experiments, 10 × 10^4^ cells/well were seeded in 6-well plates in GM for 24 h or until 75% confluency. After reaching 75% confluency, GM was removed, and each well was washed with 2 mL of PBS-1× to remove any unattached cells. Then, 2 mL of DM was added to the first group as control and Si-ions at concentrations of 0.1, 0.5, and 1.0 mM Si were added to DM for the other three groups (DM + 0.1 mM Si, DM + 0.5 mM Si, and DM + 1.0 mM Si). DM was collected for biomarker expression assays and new DM with the specified Si-ion concentration was added to each well every 48 h. After each time point, cells were fixed and immunostained following previously published protocols [74,75]. Briefly, DM was removed, and cells were washed 3 times with PBS-1×, then fixed with 2 mL of neutral buffered formalin for 7–8 min. Cells were washed again and permeabilized with 2 mL of 0.1% triton X-100 in PBS for 10 min. Then, cells were stained with 20 µL/mL of conjugated MHC antibody in 1× TBST (PBS with 0.1% Triton and 0.1% Tween) for 45 min and counterstained with DAPI (1 ug/µL). Finally, cells were washed with 2 mL/well of PBS-1× for 5 min, 3 times and fluorescence images were captured at 10× or 20× using the Leica DMi8 Inverted Fluorescence Microscope.

Fusion index [73] (FI), area covered by myotubes (%), and total number of nuclei per fluorescent image were counted to quantify myogenic differentiation of C2C12 cells. FI is defined as: the total number of nuclei within the MHC-expressing multinucleated myotubes divided by the total number of nuclei in the FOV × 100 [73]. For all experiments, three FOVs in each well were randomly selected for imaging, then ImageJ was used for cell counting and myotube area calculation.

#### 4.4.2. Different H_2_O_2_ Concentrations on C2C12 Cell Functional Capacity

To study the C2C12 cell viability under different concentrations of H_2_O_2_ as a source of ROS, the first experiment was performed to determine the minimal concentration of H_2_O_2_ that can be used to induce significant cell damage. Five concentrations of H_2_O_2_ (0.2, 0.4, 0.6, 0.8, and 1.0 mM) were used to compare the effect of H_2_O_2_ to the normal GM as control. C2C12 cell viability experiment was performed by seeding 2.5 × 10^4^ cells/well in a 24-well plate [73,74]. Then, cells were provided with 500 µL of GM as control, and GM + 0.2 mM H_2_O_2_, GM + 0.4 mM H_2_O_2_, GM + 0.6 mM H_2_O_2_, GM + 0.8 mM H_2_O_2_, and GM + 1.0 mM H_2_O_2_ as the experimental H_2_O_2_ groups. Cells were incubated for 6 and 24 h. At each time point, 12 wells (*n* = 3 each group) were used for the MTS assay and the other 12 wells (*n* = 3 each group) for the live/dead assay, as previously mentioned. Fluorescent images displaying green (live cells) and red (dead cells) were used to count the total number of live cells per FOV using ImageJ.

Based on the above cell viability experiment under ROS conditions, a concentration of 0.4 mM H_2_O_2_ was used to study the effect of Si-ions on C2C12 cell viability under toxic oxidative stress. This experiment was performed to determine the optimal Si-ions concentration for preserving C2C12 cells under toxic oxidative stress. 2.5 × 10^4^ cells/well were seeded in a 24-well plate and provided with 500 µL of specified media for each group (*n* = 3). GM was used as the positive control, GM + 0.4 mM H_2_O_2_ was used as the negative control, and four different Si-ion concentrations were used as following: GM + 0.4 mM H_2_O_2_ + 0.1 mM Si, GM + 0.4 mM H_2_O_2_ + 0.5 mM Si, GM + 0.4 mM H_2_O_2_ + 1.0 mM Si, and GM + 0.4 mM H_2_O_2_ + 2.0 mM Si. All plates were incubated for 6 and 24 h. After each time point MTS and live/dead assays were performed as previously mentioned. Fluorescence images (10× FOV) of stained C2C12 cells were taken to observe the cells after exposure to H_2_O_2_ and H_2_O_2_ + Si for 6 and 24 h.

After determining the optimal Si-ions concentration that can mitigate the effect of oxidative damage on C2C12 cells from the previous experiment, cells were cultured in media with the effective H_2_O_2_ and optimal Si-ion concentrations. In a 6-well plate, 10 × 10^4^ cells/well were cultured for 24 h or until 75% confluency. After reaching 75% confluency, GM was removed, and each well was washed with 2 mL of PBS-1× to remove any unattached cells. Then, normal DM was used as a positive control compared to 0.5 mM Si, 0.4 mM H_2_O_2_, and 0.4 mM H_2_O_2_ + 0.5 mM Si-ions. Every 48 h DM was collected from each well for biomarker expression assays and new DM was added to each well as specified. After 4 days of differentiation, cells were fixed and immunostained and counterstained using MHC and DAPI and fluorescent images were captured at 10× using a Leica DMi8 inverted fluorescence microscope. FI and area covered by myotubes were determined using ImageJ as follows. First, cell count was determined using microscope software and the cell batch counter plug in to count any clustered nuclei. The cell batch counter plug in was then used to count nuclei found within myotubes to determine the FI. The image was then converted to 8-bit, made binary, and converted to mask before determining the area covered by myotubes. The image threshold was converted to black and white, with green fluorescence set to display as black. Finally, the black regions of the image were measured to determine the percent area covered by myotubes.

### 4.5. Scratch-Wound Healing Assay Using Live Imaging Microscopy (LIM)

In this study, we have developed a scratch-wound healing assay to quantify the cells’ migration rate using dynamic live imaging microscopy (DLIM) to monitor the cells’ migration throughout scratch/wound healing. Briefly, we compared the cell migration under normal conditions using GM, oxidative stress using 0.4, 0.6, and 0.8 mM H_2_O_2_, Si supplements using 0.1, 0.5-, and 1.0-mM Si-ions, and 0.4 mM H_2_O_2_ + 1.0 mM Si, (*n* = 3 and triplicated tests). The experiments were performed by seeding 5 × 10^4^ cells/well in a 12-well plate with 1 mL GM for 24 h until full confluency. Then, a 200 µL sterile pipette tip was used to scratch the cells to introduce a thin wound, the media was removed, and each well was washed twice with PBS-1×. Then, 2 mL fresh conditioned media was added according to each group as described above. The plate was placed inside the sterile chamber of a Keyence BZX-710 fluorescence live imaging microscope (Keyence Corporation of America, IL, USA) and provided with 5% CO_2_ at 37 °C. This microscope allowed for high throughput, fully automated cell imaging at the exact same position on each well. Phase contrast images were captured every 2 min over 48 h at 10× magnification (Resolution 0.7 pixels/μm and Z plane separation = 4 µm pitch). All images, around 700 for each sample, were used to measure the wound area using the Wound Healing ImageJ software plugin [48,76] and ImageJ WH_NJ macro [77]. ImageJ software (1.52a: Wayne Rasband, National Institutes of Health, USA) was used for image processing applications, including stack projection and wound healing measurements. Keyence BZ-X Analyzer Software v.1.3.1.1. was used for avi file generation and pseudo-coloring. Linear plots of wound area verses time were extracted, and the cells’ migration rate (mm^2^/h) was calculated using the slope function.

### 4.6. Quantitative Real-Time Polymerase Chain Reaction (qRT-PCR)

QRT-PCR was used to detect changes in myogenic gene expression of C2C12 cells under normal conditions compared to treatment with Si-ions over 5 days. A cell differentiation experiment was performed as mentioned above (Section 4.4.1). Three groups of DM with Si-ions (DM + 0.1 mM Si, DM + 0.5 mM Si, and DM + 1.0 mM Si) were compared to the normal DM as a positive control. Cells were differentiated for 1, 3, and 5 days and GM were collected and stored at −20 °C for further evaluation. After each time point, cells were lysed, RNA was extracted using RNeasy Mini Kit (Qiagen, CA, USA) and converted to cDNA (Reverse Transcription System, Promega, Madison, WI, USA) according to manufacturer guidelines and our previously published protocol [78]. All groups were tested for MyoD (accession no. NM-010866.2) and MyoG (accession no. NM-031189.2) gene expression using glyceraldehyde 3-phosphate dehydrogenase (GAPDH, accession no. NM-008084.2) as the internal housekeeping gene (Applied biosystems, CA, USA). Quantification of relative gene expression was performed using the delta−delta CT (Δ−Δ-CT) method and expressed as the fold difference [78].

### 4.7. Biomarker Expression

DM was collected during the cell differentiation experiment over 5 days and stored at −20 °C. Later, collected media was analyzed for myokine expression. A mouse NRTN ELISA kit (Biomatik, DE, USA) was used to determine the concentration of NRTN secreted during C2C12 differentiation. NRTN is a key component in peroxisome-proliferator-activated receptor gamma coactivator-1α (PGC-1α) mediated neurite recruitment by skeletal muscle. NRTN ELISA kit was used according to manufacturer guidelines. NRTN-specific antibody was precoated onto the microplate in the kit, and standards and samples were pipetted into the wells to allow any present NRTN to bond with the immobilized antibody. Then, biotin-conjugated antibody specific for NRTN was added after removing any unbound molecules. After washing the plate with a wash buffer solution, avidin conjugated horseradish peroxidase (HRP) was added followed by another wash to remove any unbound avidin reagent. Then TMB substrate solution was added to each well and the plate was incubated for 15–30 min as previously described while protected from light. After 20 min, the stop solution was added, and optical density was determined within 5 min using a microplate reader set to 450 nm. Protein expression was determined based on a polynomial regression of the optical density and protein concentration of the provided standards using OriginPro 8.5 software.

The collected media was further analyzed for any markers of aminobutyric acid expression during C2C12 differentiation. Analysis was performed for all aminobutyric acid isomers, including α-aminobutyric acid (AABA), β-aminobutyric acid (BABA), and GABA. Aminobutyric acid concentrations were determined using LC-MS/MS analysis, performed on a Shimadzu LCMS-8050 triple quadrupole mass spectrometer (Shimadzu Scientific Instruments Inc., Tokyo, Japan) at the Shimadzu Center at the University of Texas at Arlington. Analysis was performed based on a previously published LC-MS/MS method that enables baseline separation with sensitive detection of aminobutyric acid isomers in minimal amounts of biological fluid samples developed by the Brotto Research Laboratory [79].

### 4.8. Statistical Data Analysis and Reporting

Immunofluorescent images were analyzed for cell number, myotube area, and FI using ImageJ. Wound Healing ImageJ software plugin [48,76] and ImageJ WH_NJ macro [77] were used for cell number, myotube area, and FI analysis. Quantitative qRT-PCR and ELISA data was evaluated using the Δ−Δ-CT method and analyzed using OriginPro 8.5 software (OriginLab Corporation, Northampton, MA, USA). The results for qRT-PCR are reported as fold differences relative to a GAPDH control. ELISA results are reported as protein concentration (ng/mL) derived by obtaining the optical densities of a set of predetermined standards then performing a standard curve regression analysis and using the resultant equation to derive the protein concentration of experimental groups from the measured optical density.

OriginPro 8.5 software was used for all graphs and statistical analysis. Statistical data are presented by individual data points and a horizontal line indicating the average of each group. Bar graphs display group means and standard deviations. One-way ANOVA followed by Tukey’s post hoc was used between group comparisons. Due to the difference in variance between the groups, following a One-way ANOVA, Tukey’s post hoc analysis was conducted to compare differences between individual groups. *p* < 0.05 was considered as statistical significance and * represents *p* < 0.05, ** for *p* < 0.01, and *** for *p* < 0.001. FI calculations were conducted blindly by the operator.

## 5. Conclusions

In this study, we explored the effect of Si-ions on C2C12 skeletal muscle cell myogenesis, such as proliferation, migration, differentiation, and myogenic biomarker expression to gain insight into its role on myogenesis during the early stages of muscle regeneration. In vitro studies indicated the addition of 0.1 mM Si-ions to media significantly increased cells’ viability, proliferation, migration, and myotube formation compared to control. Additionally, Si-ions significantly increased MyoG and MyoD gene expression within 5 days of C2C12 myoblast differentiation. Si-ions attenuated the toxic effects of H_2_O_2_ within 24 h resulting in increased cells’ viability and differentiation compared to the 0.4 mM H_2_O_2_ group. Also, 1.0 mM Si-ions increased the cells’ migration rate with a significant decrease in the scratch healing time to achieve faster healing under oxidative stress conditions. The observed upregulation of NRF-2 and SOD-1 in the 0.5 mM Si + 0.4 mM H_2_O_2_ group combined with the lack of a significant difference in expression for groups containing only Si-ions or only H_2_O_2_ suggest that Si-ions at optimal concentrations may enhance ROS metabolism and clearance in C2C12 cells under oxidative stress. These results indicate that ionic Si may have antioxidant and stimulatory effects on muscle tissue to promote skeletal muscle repair. Thus, this study provides novel evidence of the potential role of Si-ions that influence the cellular response during the myogenesis process and pave the way for designing Si-containing biomaterials with desirable Si-ions release for muscle tissue regeneration applications along with muscle drug delivery systems for degenerative disease conditions.

## Figures and Tables

**Figure 1 ijms-22-00497-f001:**
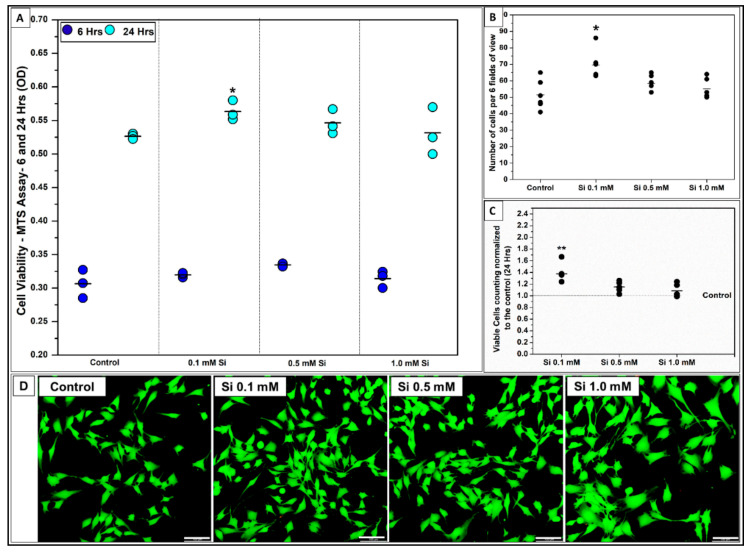
Effect of silicon ions on C2C12 myoblast cell viability. (**A**) Data distribution graph present the cell viability results using MTS-assay after 6 and 24 h of C2C12 cell culturing in growth media with three different concentration of Si^4+^ (0.1, 0.5, and 1.0 mM). 0.1 mM of Si^4+^ into the growth media significantly increases the cell viability after 24 h compared to the control (* *p* < 0.05, *n* = 3 per group). (**B**) Data distribution graph presents the number of cells per 6 fields of view confirming that 0.1 mM of Si^4+^ significantly increase the number of cells after 24 h. (**C**) Bar graph shows the number of viable cells normalized to the control after 24 h, (** *p* < 0.01). (**D**) Fluorescent pictures (20× view) of C2C12 cells stained with live/dead assay kit shows enhancement on cell viability after being exposed to ionic silicon. (—) in the graphs represents the mean and the scale bar in the fluorescent pictures is 100 µm.

**Figure 2 ijms-22-00497-f002:**
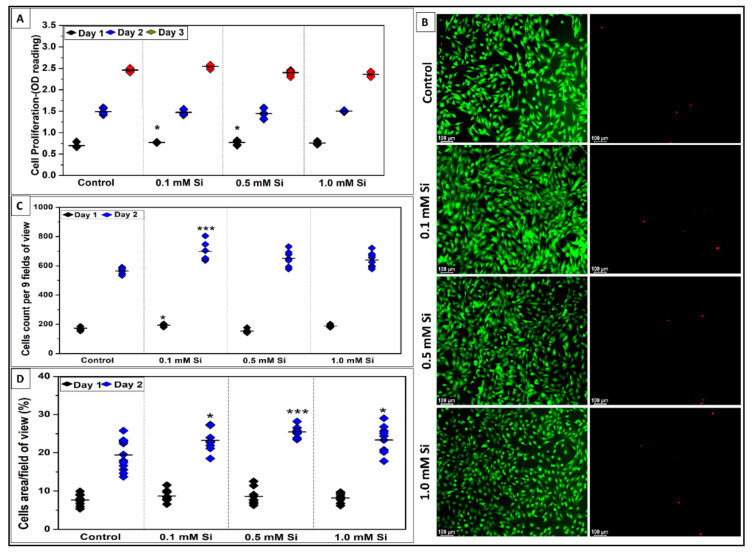
Effect of Si-ions concentration on C2C12 cells proliferation for 3 days. (**A**) Data distribution graph shows the MTS-assay results of C2C12 cell proliferation with 3 different concentration of silicon ions for 1, 2, and 3 days. 0.1 mM Si significantly increased the number of cells at day 1 and 3, (* *p* < 0.05). (**B**) Fluorescent pictures (10× view) of C2C12 cells stained with live/dead assay kit shows the live (green) and dead (red) cells after 2 days of proliferation. Almost no dead cells were observed for the control and silicon ions groups. (**C**) Data distribution graph presents the number of cells per 9 fields of view counted from the fluorescent pictures. The cell counts confirmed that 0.1, 0.5, and 1.0 mM Si-ions significantly increased the total number of cells by day 2 of proliferation. (**D**) Cell area percentage confirmed that Si-ions significantly increased the area covered by C2C12. (—) in the graphs represents the mean and the scale bar on the pictures is 100 µm, (*** *p* < 0.001, *n* = 3 per group).

**Figure 3 ijms-22-00497-f003:**
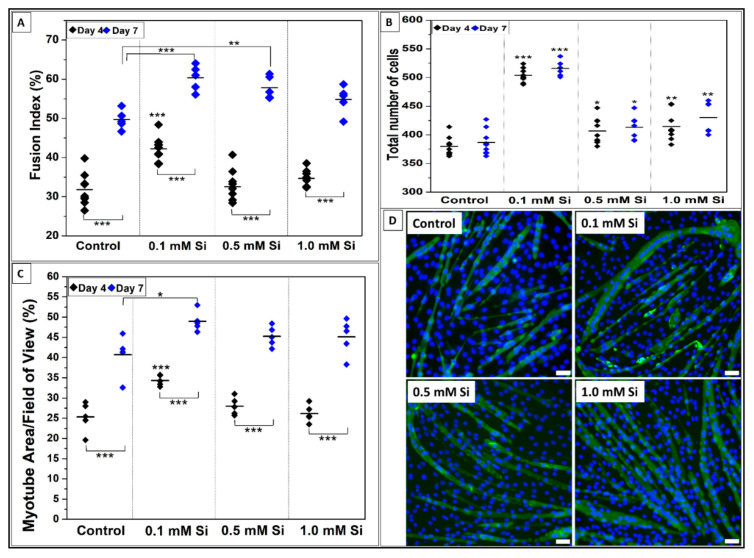
Effect of Si-ions concentration on C2C12 cells after 4 and 7 days of differentiation. (**A**) Fusion index (FI) indicates that 0.1 mM of silicon significantly increases the FI compared to the control after 4 days. Additionally, 0.1, 0.5, and 1.0 mM of silicon significantly increases the FI compared to the control after 7 days (*** *p* < 0.001, ** *p* < 0.01, * *p* < 0.05). FI was significantly increased from day 4 to day 7 for all samples (*** *p* < 0.001). (**B**) Total number of cells counted from 9 fields of view of DAPI stained nuclei, all used silicon concentration significantly increased the total number of cells at day 4 and 7 compared to the control and 0.1 mM was the optimal concentration with *** *p* < 0.001 significance. There was no significant difference between the total number of cells from day 4 to 7. (**C**) Area covered by myotubes (%) also indicated that all used silicon concentrations increase the total area of myotubes per field of view compared to the control and 0.1 mM Si showed high significance at day 4 and 7. Also, the myotube area showed a significant increase from day 4 to 7 for all samples and the control. (**D**) represents fluorescence images of DAPI-stained nuclei (blue) and myosin heavy chain antibody (MHC, green)-stained myocytes/myotubes of C2C12 myoblasts on the tissue culture plate (TCP) as a control and the three different Si-ions concentrations (0.1 mM, 0.5 mM, and 1.0 mM) after 4 days of differentiation. Scale bar is 50 µm and *** *p* < 0.001, ** *p* < 0.01, * *p* < 0.05.

**Figure 4 ijms-22-00497-f004:**
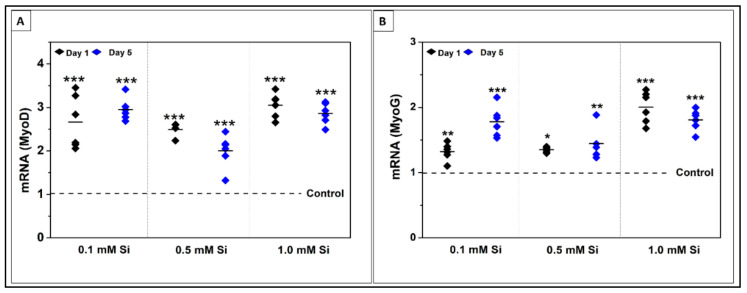
Myogenic determination protein (MyoD) and Myogenin (MyoG) genes expression at 1 and 5 days of C2C12 cell differentiation. (**A**) Si-ions increased the expression of MyoD at early stage of differentiation (Day 1); 0.1 and 1.0 mM of Si-ions significantly increased MyoD expression at day 1 and 5 (* *p* < 0.05, ** *p* < 0.01, *** *p* < 0.001). (**B**) MyoG expressed at a later stage marking the commitment to differentiation, 1.0 mM Si-ions significantly increased MyoG expression by 2-folds at day 1. By day 5, both 0.1 and 1.0 mM of Si-ions increased MyoG expression almost 2-fols compared to the control.

**Figure 5 ijms-22-00497-f005:**
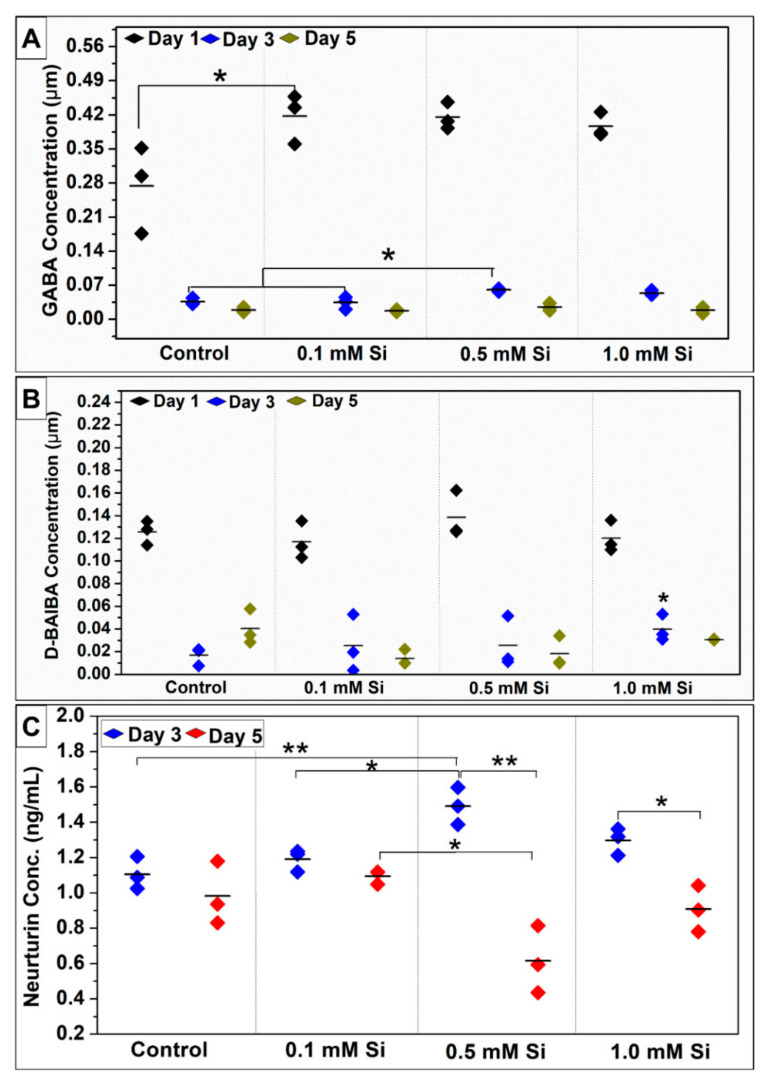
Effect of Silicon-ions on ϒ-aminobutyric acid (GABA), D-Beta aminoisobutyric acid (D-BAIBA) concentration in µM, and Neurturin (ng/mL) expressed by C2C12 during 5 days of differentiation. (**A**) GABA concentration increased by adding Si-ions to the differentiation media. (**B**) The concentration of D-BAIBA increased by adding 0.5 mM of Si to the differentiation media at day 1, but no significant difference was observed compared to the control. (**C**) Neurturin expression in media (ng/mL) during C2C12 myoblast cell differentiation for 3 and 5 days. All silicon-ion concentrations increased the neurturin expression, but the 0.5 mM of silicon-ions significantly increased the neurturin expression compared to the control at day 3 of differentiation (**p* < 0.05 and ** *p* < 0.01).

**Figure 6 ijms-22-00497-f006:**
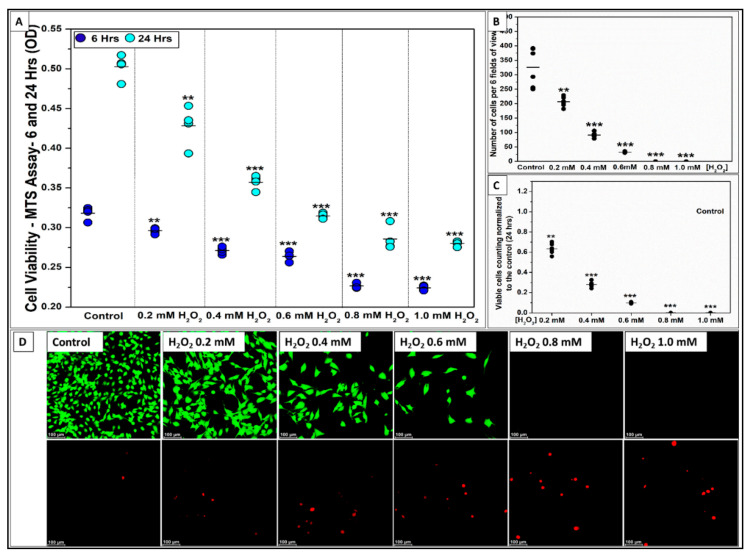
Effect of Hydrogen peroxide (H_2_O_2_, ROS-source) on C2C12 myoblast cell viability. (**A**) Data distribution graph presents the cell viability results using MTS assay after 6 and 24 h of C2C12 cell culturing in growth media with five different concentration of H_2_O_2_ (0.2, 0.4, 0.6, 0.8, and 1.0 mM). 0.4 mM of H_2_O_2_ into the growth media significantly decreases the cell viability after 6 and 24 h compared to the control (*** *p* < 0.001, ** *p* < 0.01, *n* = 4 per group). (**B**) Data distribution graph presents the number of cells per 6 fields of view confirming the same results from the MTS-assay. (**C**) Bar graph shows the number of viable cells normalized to the control after 24 h, 0.2 mM of H_2_O_2_ decreased the number of viable cells to 63% and the 0.4 mM H_2_O_2_ to ~30% viable cells compared to the control 100% (0 mM H_2_O_2_). (**D**) Fluorescent pictures (20× view) of C2C12 cells stained with live/dead assay kit shows the live (green) and dead (red) cells after being exposed to H_2_O_2_ for 24 h. No viable cells were observed with 0.8 and 1.0 mM of H_2_O_2_. (—) in the graphs represents the mean and the scale bar on the fluorescent pictures is 100 µm.

**Figure 7 ijms-22-00497-f007:**
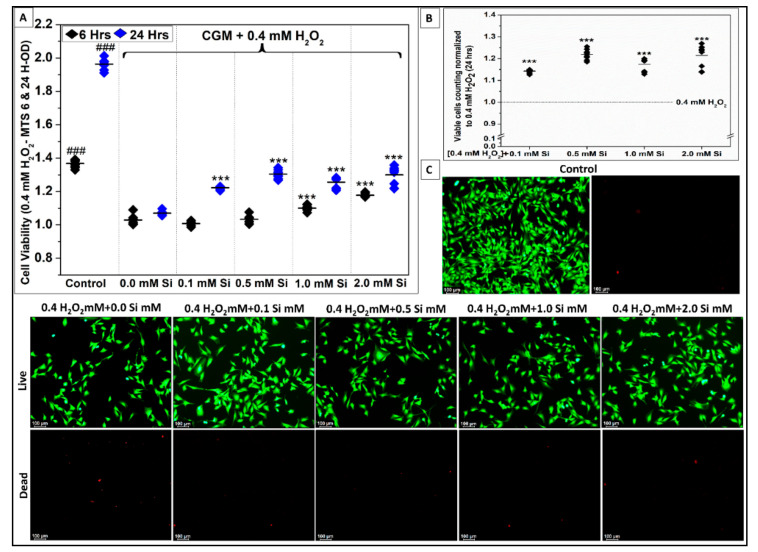
Silicon ions attenuate the effect of toxic oxidative stress (0.4 mM H_2_O_2_) on C2C12 myoblast cells. (**A**) Data distribution graph presents the cell viability results using MTS assay after 6 and 24 h of C2C12 cell culturing in complete growth media (CGM, Control), supplied with 0.4 mM H_2_O_2_, (test groups) with 0.0, 0.1, 0.5, 1.0 mM Si-ions. 0.0 mM Si indicates that 0.4 mM of H_2_O_2_ into the growth media significantly decreases the cell viability after 6 and 24 h compared to the control (*** *p* < 0.001, *n* = 4 per group), addition of 0.5–1.0 mM of Si into the growth media under ROS significantly enhances the cell viability compared to 0.4 mM H_2_O_2_ without Si (*p* < 0.05, *n* = 4 per group). (**B**) Data distribution graph presents the number of cells normalized to 0.4 mM H_2_O_2_ without Si confirming the same results from the MTS-assay. (**C**) Fluorescent pictures (10× view) of C2C12 cells stained with calcine Am stain showing cells after being exposed to H_2_O_2_ for 24 h. (^___^) in the graphs represents the mean and the scale bar on the fluorescent pictures is 100 µm. ### *p* < 0.001.

**Figure 8 ijms-22-00497-f008:**
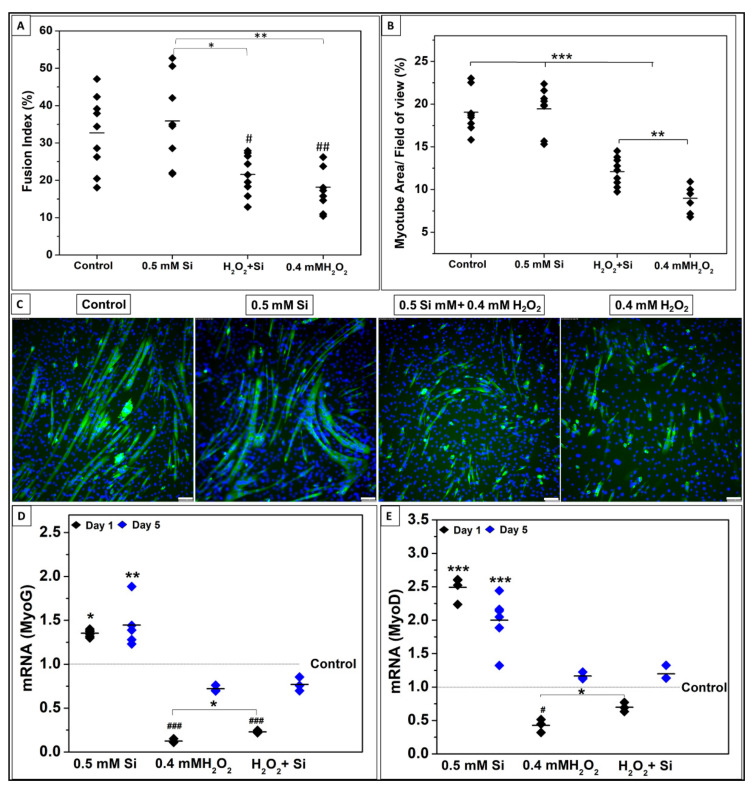
Effects of Si-ions concentration on C2C12 myoblast cells after 4 days of differentiation under oxidative stress (0.4 mM H_2_O_2_). (**A**) Data distribution graph shows the Fusion index (FI) after 4 days of differentiation comparing the differentiation under 0.4 mM H_2_O_2_ to 0.4 mM H_2_O_2_ + 0.5 mM Si and the control. Adding 0.5 mM Si increases the FI compared to 0.4 mM H_2_O_2_, but no significance deference was observed. (**B**) Area covered by myotubes (%) indicated that addition of 0.5 mM Si increase the total area of myotubes compared to 0.4 mM H_2_O_2_ by reliving the cells from the toxic oxidative stress. (**C**) Fluorescence images of DAPI-stained nuclei (blue) and myosin heavy chain antibody (MHC, green)-stained myocytes/myotubes of C2C12 myoblasts on the tissue culture plate (TCP) as a control, 0.5 mM Si-ions, 0.4 mM H_2_O_2_ + 0.5 mM Si, and 0.4 mM H_2_O_2_ after 4 days of differentiation, Scale bar is 100 µm, 10-X magnification. (**D**,**E**) Myogenic determination protein (MyoD) and Myogenin (MyoG) genes expression at 1 and 5 days of C2C12 cell differentiation under ROS conditions. The * was used to show the significant increase (* *p* < 0.05, ** *p* < 0.01, *** *p* < 0.001) while # shows the significant decrease (# *p* < 0.05, ## *p* < 0.01, ### *p* < 0.001).

**Figure 9 ijms-22-00497-f009:**
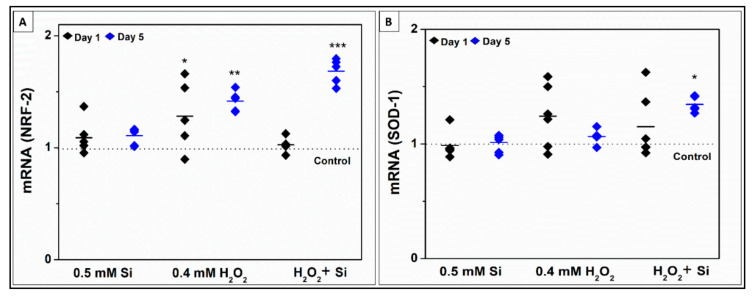
Si ions enhance antioxidant marker expression in the presence of toxic oxidative stress. (**A**) Relative concentration of NRF-2 mRNA expressed by C2C12 skeletal muscle cells differentiated in media containing 0.5 mM silicon ions only, 0.4 mM H_2_O_2_ only, and media containing 0.5 mM Si Ions + 0.4 mM H_2_O_2_ for 1 and 5 days. NRF-2 expression was significantly upregulated in the 0.4 mM H_2_O_2_ group at days 1 (1.28, *p* = 0.03) and 5 (1.41, *p* = 0.008) compared to the control, and in the H_2_O_2_ + Si group at day 5 (1.7, *p* = 0.00002). (**B**) Relative concentration of SOD-1 mRNA expressed by C2C12 skeletal muscle cells differentiated in media containing 0.5 mM silicon ions only, 0.4 mM H_2_O_2_ only, and media containing 0.5 mM Si Ions + 0.4 mM H_2_O_2_ for 1 and 5 days. SOD-1 expression was significantly upregulated in the H_2_O_2_ + Si group compared to the control at day 5 (1.35, *p* = 0.04), (* *p* < 0.05, ** *p* < 0.01, *** *p* < 0.001) and no other significant differences were observed.

**Figure 10 ijms-22-00497-f010:**
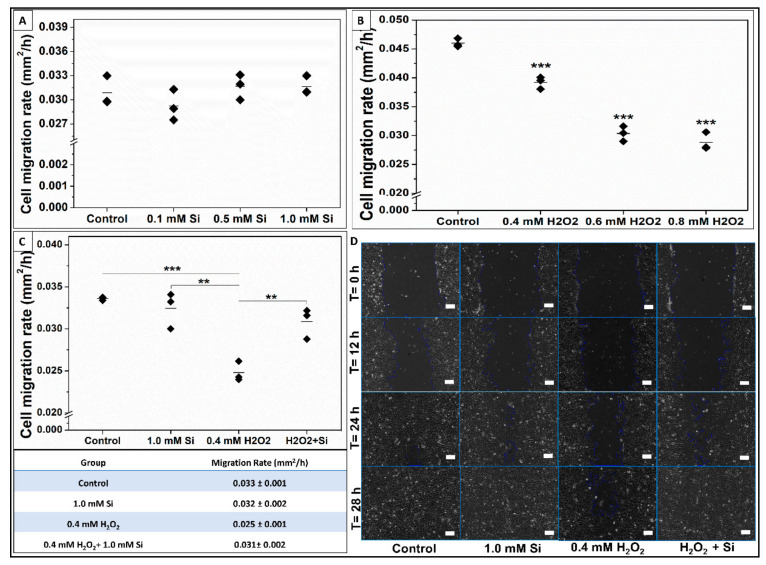
Si-ions significantly enhance muscle wound healing by increasing the cells’ migration rate under toxic oxidative stress condition. (**A**) Data presents the cell migration rate (mm^2^/h) of C2C12 myoblast cells under normal control and three concentrations of Si-ions. Growth media with 0.5–1.0 mM Si-ions enhances the cells’ migration rate. (**B**) H_2_O_2_ significantly decreases the cells’ migration rate under the used concentrations (0.4, 0.6, and 0.8 mM H_2_O_2_) (*** *p* < 0.001). (**C**) Si-ions attenuate the toxic effect of H_2_O_2_ and significantly increase the cells’ migration rate under toxic oxidative stress condition. (**D**) Bright field images (10×, scale bar = 100 μm) show the wound/scratch area at different time points (0, 12, 24, and 28 h), (** *p* < 0.01, *** *p* < 0.001).

## Data Availability

The data presented in this study are available on request from the corresponding author.

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
