# Peer review of "Ionic Silicon Protects Oxidative Damage and Promotes Skeletal Muscle Cell Regeneration"

_ijms, 2021, doi:10.3390/ijms22020497_

Round 1

Reviewer 1 Report

The paper by Awad et al. study how silicon ions protects skeletal muscle from oxidative damage. The question is of interest and provides a new avenue for future therapeutics of muscle dystrophies and to prevent muscle waste.

I have some concerns:

Introduction:

The authors describe NRTN, GABA and BAIBA as myokines. Howeer, the most study myokine is IL-6. Why haven´t you study IL-6? Indeed, it has been associated with at least muscle sarcopenia.

Lines 74-77. This is not accurate. In fact, antioxidants are not always good for muscles for instance, during exercise.  Please provide some examples (references) in order to get the context.   

Results:

 Figure 5, Why NRT was not measured in day 1? I suggest to use the same colors to denote each time-point.

L369 I do not believe that the increase in NFR2 is due to SI treatment as the H2O2 dose used (0.4 mM) show a similar increase.

However, the effect on SOD1 is clearly dependent upon SI treatment. In this regard, some muscle dystrophies are related to SOD deficiencies it would be interesting to add some discussion for future preclinical studies.

Please, ensure that all the acronyms are previously defined.

Author Response

Response to reviewer comments for IJMS

We wish to thank reviewers for their helpful comments to help improve the manuscript for further consideration in IJMS. Please review our point-by-point responses below and our uploaded corrected manuscript with track changes. Thank you all for your time again in reviewing our manuscript. We are very grateful.

Reviewer 1

The paper by Awad et al. study how silicon ions protects skeletal muscle from oxidative damage. The question is of interest and provides a new avenue for future therapeutics of muscle dystrophies and to prevent muscle waste.

Response: We thank the reviewer for their kind comment. We have endeavored to respond to the questions below to help strengthen the manuscript.

1. The authors describe NRTN, GABA and BAIBA as myokines. However, the most study myokine is IL-6. Why did you not study IL-6? Indeed, it has been associated with at least muscle sarcopenia.

Response: We totally agree with the reviewer that the IL-6 is one of the most studied myokines, but it is very complex due to its pleiotropic actions. Although IL-6 increases in response to exercise and fasting, impaired in sarcopenia, it also can act as a pro-inflammatory cytokine and an anti-inflammatory myokine. Thus, IL-6 levels vary substantially depending on the model and timing of measurement. Based on that, we decided to study these three myokines instead, (NRTN, GABA, and BAIBA) to capture a better-defined image of Si effect on these myokines within the context of myogenesis. NRTN was selected due to its role in motor nerve recruitment and neuromuscular junction remodeling. GABA is the chief inhibitory neurotransmitter in the central nervous system and very important for muscle tonus, and BAIBA can protect osteocytes from toxic ROS and prevent both bone and muscle loss due to unloading. We clarified each of these points in the Revised Manuscript in the introduction section.

2. Lines 74-77. This is not accurate. In fact, antioxidants are not always good for muscles for instance, during exercise. Please provide some examples (references) to get the context.  

Response: We thank the reviewer for this suggestion, the following paragraph was inserted in the introduction section (Page 2, line 78-81): (Although, experimental evidence has shown that antioxidant supplementations have no effect on muscle function during and after exercise [33], dietary antioxidants have been demonstrated to improve antioxidant enzyme expression and subsequent muscle repair and function in muscle injuries [34,35]. Other investigations have explored administering low doses of antioxidants, leading to reduced oxidative stress, cell cycle and migration stimulation, and enhanced myogenesis [28,36]).

3. Figure 5, Why NRT was not measured in day 1? I suggest using the same colors to denote each time-point.

Response: We thank the reviewer for this suggestion. Figure 5 was updated, and the same color was used to denote each specific time point. Regarding the specific suggestion of measuring NRTN at day 1, our main goal in this situation was to link with the progression of myogenic differentiation at Day 3 and later at the peak of the process at Day 5.

4. L369 I do not believe that the increase in NFR2 is due to Si treatment as the H2O2 dose used (0.4 mM) show a similar increase.

Response: We thank the reviewer for this note, we made it more clear and updated the text as follows:

In the results section (Page 15, line 387-391): “At normal condition, Si treatment did not show any significant effect on NRF-2 or SOD-1 expression compared to the control, while 0.4 mM H2O2 has increased the SOD-1 and NRF-2 expression with a significant increase only in NRF-2 expression at day 1 and 5 (Figure 9-A). For the combined group (0.5mM Si + 0.4mM H2O2), there was a significant increase in NRF-2 (1.7±0.1, p=0.00002) and SOD-1 (1.35±0.06, p=0.04) at day 5 compared to the control.” 

In the discussion section (Page 18, line 537-542) “Additionally, the results of qRT-PCR assays suggest that Si may upregulate the expression of SOD-1 when in the presence of oxidative stress, however, the same effect was not observed under normal conditions or oxidative stress conditions in the absence of Si-ions. Although 0.4 mM H2O2 group presented a significant increase in NRF-2 expression (1.4 fold change), the expression was more pronounced under Si treatment with 1.7 fold change.”

5. However, the effect on SOD1 is clearly dependent upon Si treatment. In this regard, some muscle dystrophies are related to SOD deficiencies it would be interesting to add some discussion for future preclinical studies.

Response: We thank the reviewer for this suggestion, the following paragraph was inserted (discussion section, Page 19, line 543-548): “H2O2 can react with metal ions in the cells to produce hydroxyl radical which is one of the most reactive species in the biological systems [62]. High levels of ROS with imbalanced antioxidants expression usually generates an oxidative stress that leads to degenerative changes to muscle with all of the characteristics of a muscular dystrophy [60–63]. In this regard, Si treatment could be a beneficial approach for upregulation of SOD-1 in some muscle dystrophies related SOD deficiencies.”

6. Please, ensure that all the acronyms are previously defined.

Response: We thank the reviewer for this note, we have defined all used acronyms.

Reviewer 2 Report

In the manuscript “Ionic Silicon Protects Oxidative Damage and Promotes Skeletal Muscle Cell Regeneration” the authors investigate the effect of silicon-ions on C2C12 skeletal muscle cells under normal and excessive oxidative stress conditions to determine its role on myogenesis during the early stages of muscle regeneration. I find the subject of article interesting. Methodology used is in general correct and obtained results are promising. Some points are need to be address for the sake of the manuscript value.

Major comment

In section 2.2.3 authors have describe that Si-ions enhance cell differentiation under toxic oxidative stress. Myogenic transcription factors (MyoG and MyoD) regulate the myogenic differentiation process. Therefore, it would better to add additional data on level of MyoG and MyoD expression in this section also.

Minor comments

  1. Line #18, #236, #256, #750, in vitro should be in italic.
  2. Line #26, ROS should be write in full.
  3. Line #29, ROS and VML should not be in abbreviated form in key words.
  4. Line #63, Line #26, ROS should be write in full, and then abbreviate.
  5. Line #191 and #192, ref 22, 21 should be in bracket.
  6. Line #210 and #214, check typing error for ‘a’, and ‘b’. It should be ‘A’ and ‘B’.
  7. Line #284 and #285, correct (Figure 6b) and Figure 6c as (Figure 6B) and Figure 6C.  
  8. The typo errors throughout the manuscript should be corrected.
  9. References are not prepared as per the authors guidelines.

Author Response

Thank You. 

Round 2

Reviewer 2 Report

No comments